# Fabrication of Polypyrrole Hollow Nanospheres by Hard-Template Method for Supercapacitor Electrode Material

**DOI:** 10.3390/molecules29102331

**Published:** 2024-05-15

**Authors:** Renzhou Hong, Xijun Zhao, Rongyu Lu, Meng You, Xiaofang Chen, Xiaoming Yang

**Affiliations:** State and Local Joint Engineering Laboratory for Novel Functional Polymeric Materials, Jiangsu Engineering Laboratory of Novel Functional Polymeric Materials, Suzhou Key Laboratory of Macromolecular Design and Precision Synthesis, Department of Polymer Science and Engineering, College of Chemistry, Chemical Engineering and Materials Science, Soochow University, Suzhou 215123, China; 20214209184@stu.suda.edu.cn (R.H.); 20214209280@stu.suda.edu.cn (X.Z.); 20224209228@stu.suda.edu.cn (R.L.); 20234209022@stu.suda.edu.cn (M.Y.)

**Keywords:** hollow polypyrrole spheres, energy storage, nanostructure, hard template

## Abstract

Conducting polymers like polypyrrole, polyaniline, and polythiophene with nanostructures offers several advantages, such as high conductivity, a conjugated structure, and a large surface area, making them highly desirable for energy storage applications. However, the direct synthesis of conducting polymers with nanostructures poses a challenge. In this study, we employed a hard template method to fabricate polystyrene@polypyrrole (PS@PPy) core–shell nanoparticles. It is important to note that PS itself is a nonconductive material that hinders electron and ion transport, compromising the desired electrochemical properties. To overcome this limitation, the PS cores were removed using organic solvents to create hollow PPy nanospheres. We investigated six different organic solvents (cyclohexane, toluene, tetrahydrofuran, chloroform, acetone, and N,N-dimethylformamide (DMF)) for etching the PS cores. The resulting hollow PPy nanospheres showed various nanostructures, including intact, hollow, buckling, and collapsed structures, depending on the thickness of the PPy shell and the organic solvent used. PPy nanospheres synthesized with DMF demonstrated superior electrochemical properties compared to those prepared with other solvents, attributed to their highly effective PS removal efficiency, increased specific surface area, and improved charge transport efficiency. The specific capacitances of PPy nanospheres treated with DMF were as high as 350 F/g at 1 A/g. And the corresponding symmetric supercapacitor demonstrated a maximum energy density of 40 Wh/kg at a power density of 490 W/kg. These findings provide new insights into the synthesis method and energy storage mechanisms of PPy nanoparticles.

## 1. Introduction

Supercapacitors are energy storage devices with higher power density and cyclability than batteries [1]. However, their lower energy density limits practical applications, especially in portable devices. Recent research has focused on increasing supercapacitor energy density without compromising power density. Supercapacitors are typically divided into electrochemical double-layer capacitors (EDLCs) and pseudocapacitors [2]. EDLCs store energy through charge adsorption/desorption at the electric–ionic conductor interface using high surface area carbon materials. Pseudocapacitors offer higher energy density by incorporating reversible redox reactions. Developing high-energy and power density pseudocapacitors with rational electrode structures is a practical alternative to EDLCs [3].

Polypyrrole (PPy) has been extensively studied as an active material for pseudocapacitors due to its high conductivity in doped states (~100–10,000 S/m), cost-effectiveness, ease of large-scale fabrication, and better environmental stability. It has found widespread applications in the field of supercapacitors [4,5,6,7,8]. Efforts have been made to enhance the performance of PPy-based supercapacitors. PPy with nanostructures, such as spheres, nanotubes, nanowires, 2D sheets, and 3D porous films, has demonstrated superior electrochemical performance due to its high surface area, conductivity, and π-conjugated structure [9,10,11]. The specific capacity of hollow PPy sphere-based supercapacitors can achieve 252 F g^−1^ at the current density of 0.5 A g^−1^ [12]. The MoS_2_/PPy/reduced graphene oxide electrode exhibits a specific capacitance of 1942 F g^−1^ at a density of 1 A g^−1^ [13]. A 3D porous PPy film electrode delivered a specific capacitance of 313.6 F g^−1^ and 98.0 mF cm^−2^ at 1.0 A g^−1^ in a three-electrode configuration and 62.5 F g^−1^ at 0.5 A g^−1^ in the symmetric capacitor device [14]. Nanostructured PPy is typically obtained through soft or hard templating methods. Soft templating, using surfactant vesicles, lacks precise control over the nanostructure of PPy [4,15]. The hard-templating method involves synthesizing templates with specific shapes, coating them with the desired materials, and selectively removing the template materials to obtain replication nanostructures [16,17]. Among various nanostructures of PPy, the PPy hollow nanospheres are a unique and traditional nanostructure that contains different cavity and shell structures, offering advantages such as high specific surface area, high porosity, large cavity volume, and excellent surface permeability. The synthesis of PPy hollow nanospheres using the hard template approach, particularly by coating commercial polystyrene (PS) spheres with thin PPy shells and selectively dissolving the PS template, has been widely reported [18,19]. However, PS itself is a nonconductive material that hinders electron and ion transport, thereby compromising the desired electrochemical properties. To overcome this limitation, the PS cores were removed using organic solvents to create hollow PPy nanospheres. Different organic solvents, such as cyclohexane (CHX), tetrahydrofuran (THF), chloroform (CHCl_3_), toluene (TOL), acetone (AC), and N,N-dimethylformamide (DMF) have been reported for PS core dissolution [20,21,22,23,24,25,26,27]. However, the dissolution process and the corresponding structure of the PPy shell, as well as its impact on the electrochemical properties, have not been extensively studied.

In this study, PS@PPy core–shell nanoparticles with various PPy thicknesses were initially synthesized using the hard template method. Subsequently, the PS cores were etched using six organic solvents (CHX, TOL, THF, CHCl_3_, AC, and DMF). The etching process and the corresponding nanostructure of the PPy shell were investigated, and the electrochemical properties of the resulting hollow PPy nanospheres were also evaluated.

## 2. Results and Discussion

### 2.1. Synthesis Process and Structural Characterization of Hollow PPy Nanospheres

Figure 1a demonstrates the formation of PS@PPy core–shell nanoparticles and corresponding hollow PPy particles using an organic solvent etching strategy. The formation mechanism of the PS@PPy core–shell nanoparticles is attributed to the electrostatic interaction between positively charged PPy and negatively charged PS particles [3,24,28,29,30].

The scanning electron microscopy (SEM) images in Figure 1b–e depict the PS and PS@PPy core–shell nanoparticles with different PPy thicknesses, which exhibit a smooth surface and consistent spherical shape. The average diameter of the PS was approximately 220 nm, with a standard deviation of 14 nm. The PS@PPy core–shell nanoparticles possess three variations: PS@PPy20, PS@PPy30, and PS@PPy40, each representing different shell thicknesses of 20 nm, 30 nm, and 40 nm, respectively. Then, the PS templates were selectively removed through chemical etching to obtain hollow PPy nanospheres.

Polystyrene (PS) has a solubility parameter of 9.1 cal^0.5^ cm^−1.5^, which allows for solubility in solvents such as CHX, TOL, THF, CHCl_3_, AC, and DMF due to similarities in their solubility parameters (Table 1). The high molecular weight (Mw) of 293,244 g/mol for the PS cores means that the dissolution process in organic solvents is complex (Appendix A), as the PS chains are entangled above its entanglement molecular weight (Me) of 16,910 g/mol [31]. When a solvent contacts PS, the solvent molecules diffuse into and swell it [31,32]. This transition from the glassy to the rubbery state of the PS chains increases chain separation. Above a critical swelling, chains disentangle and dissolve into the solvent. For PS solid cores encapsulated in a rigid PPy shell, swelling is initially restricted (Figure 1a). The absorbed solvent pressurizes the core against the shell as PS chains cannot separate further. Once the pressure exceeds a certain threshold, cracks spontaneously form in the shell, allowing rapid escape of the swollen PS chains [33]. Alternatively, the PPy shell may deform as PS chains or organic solvent molecules migrate out. The key steps are diffusion and swelling of the high-Mw PS core restricted by the PPy shell, followed by applied pressure until shell failure releases the swollen chains.

To remove the PS core and obtain hollow PPy spheres, six organic solvents (CHX, TOL, THF, CHCl_3_, AC, and DMF) were investigated for this study. Table 1 provides the assigned numbers for these solvents based on their polarity and solubility parameters. The PS@PPy particles were exposed to these solvents to dissolve the PS core. Afterward, centrifugation cycles were performed to wash the particles with the corresponding organic solvent, followed by drying in a vacuum oven. The SEM and transmission electron microscopy (TEM) images in Figure 2 and Appendix A illustrate the morphology of the PS@PPy particles after the organic solvent dissolution process.

The properties of organic solvents, such as polarity, solubility, and volatility, as well as the thickness of the PPy shell, significantly impact the morphology of hollow PPy spheres. Appendix A shows SEM and TEM images of CHX PS@PPy with varying shell thicknesses. Despite the slight dissolution of the PS cores due to their different solubility parameters compared to that of PS, the spherical morphology remains intact. SEM and TEM images of TOL PS@PPy with different thicknesses are presented in Appendix A. TOL PS@PPy20 crushes due to its thinnest shell, while TOL PS@PPy30 and TOL PS@PPy40 exhibit buckling and remain intact, with partial dissolution of the PS core. THF PS@PPy20 also has a collapsed structure, while THF PS@PPy30 and THF PS@PPy40 exhibit buckled and hollow structures, respectively, due to THF’s excellent dissolution properties (Appendix A). On the other hand, CHCl_3_ and AC, which are highly volatile, are unable to maintain their spherical structure, which results in partial dissolution of the PS cores (Appendix A). Interestingly, DMF PS@PPy20, DMF PS@PPy30, and DMF PS@PPy40 demonstrate buckling, hollow, and intact spheres, respectively. DMF exhibits the highest PS removal efficiency due to its high polarity, which is consistent with the PPy shell and facilitates the diffusion process of PS chains out of the PPy shell [34]. Consequently, further investigations focused on DMF PS@PPy, with PS@PPy serving as a control sample, to explore the morphology, structure, and relationship with the electrochemical properties of the hollow PPy spheres.

Figure 2d displays Fourier transform infrared (FTIR) spectra of pristine PPy, PS, the PS@PPy particles, and DMF PS@PPy particles to investigate the molecular structure of PPy. The PPy control sample exhibited characteristic peaks, including a wide peak at 3400 cm^−1^ (N-H stretching vibration), peaks at 1554 cm^−1^ (antisymmetric pyrrole ring vibrations) and 1454 cm^−1^ (symmetric pyrrole ring vibrations), bands at 1288 cm^−1^ (C-N stretching vibrations) and 1030 cm^−1^ (C-H deformation vibrations), a peak at 1170 cm^−1^ (pyrrole ring breathing vibration), and a peak at 910 cm^−1^ (C-H deformation vibration), consistent with previous reports confirming PPy formation [35]. All the characteristic peaks of PPy are present in the spectrum of PS@PPy, indicating PPy formation. The peaks of PPy show a minor redshift compared to those of pristine PPy, indicating the presence of delocalized π-electrons from the PPy backbone. In the case of DMF PS@PPy, most of the PS peaks overlapped with those of PPy, but weak bands at 3026 cm^−1^ and 1658 cm^−1^ (C-H and C=C stretching of phenyl rings) were observed, confirming the incomplete removal of PS. This may be due to the formation of PPy-PS hybrids during the process of pyrrole polymerization on the surface of the PS cores [24].

Thermogravimetric analysis (TGA) was also conducted under a nitrogen atmosphere to assess the thermal endurance of the PPy-based samples. Figure 2e,f display the thermal degradation profiles of PS, PPy, and the core–shell PS@PPy particles. As shown in Figure 2e, the initial decomposition temperature of PS was approximately 270 °C [27]. However, the core–shell PS@PPy sample exhibited an onset point of decomposition at 340 °C. This finding provides evidence that the PPy shell shields the PS cores, thereby enhancing the decomposition temperature. According to the TGA curves of PPy, H_2_O starts to evaporate below 150 °C, while the decomposition of PPy chains begins at approximately 200 °C. The residue weights of all the PPy samples at 800 °C indicate a high carbon yield for PPy [36].

Figure 2f shows the TGA curves of DMF PS@PPy20/30/40. The significant mass decrease observed at 340 °C is attributed to residual PS. Consequently, it can be concluded that the formation of PPy-PS hybrids during the process of pyrrole polymerization on the surface of the PS cores prevents complete dissolution of the PS cores by organic solvents [24]. The TGA results are consistent with the FTIR results; that is, the organic solvent cannot completely dissolve the PS core, while DMF exhibits the highest PS removal efficiency [24].

X-ray photoelectron spectroscopy (XPS) was employed to analyze the surface compositions of the samples (Figure 3). The C_1s_ (~284.5 eV), O_1s_ (~532 eV), Cl_2s_, Cl_2p_ (~199.5 eV), and N_1s_ (~399.8 eV) signals, originating from the PS cores and PPy shell, respectively, were viewed in the survey spectra of the core–shell PS@PPy nanoparticles and DMF PS@PPy (Figure 3a–c) [37]. The O_1s_ signal originated from the surface oxidation of PPy or weakly charge-transfer-complexed oxygen atoms [11,38]. The Cl_2s_ and Cl_2p_ signals correspond to the typical spectra of chlorine-doped PPy. The chloride originated from the polymerization process with iron chloride. The N_1s_ XPS signal (Figure 3e,f) effectively confirms the elemental identity of the prepared PPy shell, providing distinct evidence for the presence of PPy in the composition [39].

Figure 3d–f show the nitrogen (N_1s_) spectrum obtained from the XPS of all the samples. For all the samples, the nitrogen signals were divided into three peaks corresponding to =NH-, -NH-, and -NH^+^- [40]. The high binding energy tail at 400.2 eV is attributed to positively charged nitrogen (-NH^+^-). Treatment with DMF reduced the high binding energy tail intensity, and a shoulder appeared at approximately 397.9 eV attributed to imine nitrogen (-N=) [41]. By comparing the ratio of doped (non-neutral) nitrogen to total nitrogen signals, the doping degrees for PS@PPy30, DMF PS@PPy20, and DMF PS@PPy30 were determined to be 68.5%, 25.4%, and 28%, respectively (Table 2). The doping level of PS@PPy30, represented by the N^+^/N ratio, is 68.5%, significantly higher than the reported N^+^/N ratio for pristine PPy. Typically, the doping level of PPy achieved through electrostatic interactions with anions ranges from one-fourth to one-third. This indicates the enhanced doping level of PPy induced by the PS surface [42]. After DMF treatment, the oxygen content of the sample increased while the doping level decreased due to the formation of oxidized species [43]. It is concluded that organic solvent negatively impacts the PPy doping level, which may deteriorate the electrochemical properties of the PPy samples, which will be discussed in the following section.

### 2.2. Electrochemical Performance of PPy in a Three-Electrode System

The interconnections between morphology, structure, and electrochemical properties of PS@PPy treated with organic solvent were explored. Initially, cyclic voltammetry (CV) was conducted to probe the fundamental electrochemical characteristics of the PPy-based samples in a three-electrode system, employing 1 M H_2_SO_4_ electrolyte at a scan rate of 5 to 100 mV s^−1^ [44]. Figure 4a–c illustrate representative cyclic voltammetry (CV) profiles of the PPy electrodes treated with different organic solvents at a scan rate of 100 mV s^−1^. As controls, the PS@PPy20/30/40 samples were also examined. The recorded currents were normalized by dividing them by the weight of the active materials. The observed nearly rectangular shape of the CV curves (Appendix A) indicates improved capacitance and swift ion response. Among the various organic solvents, DMF PS@PPy exhibits the largest CV area, signifying superior capacitive behavior and excellent carrier transport compared to PS@PPy and PPy treated with other organic solvents. This result indicated the advantages of hollow PPy structure and highlighted the PS removement efficiency of DMF since the nonconductive PS core hinders carrier transfer.

We performed galvanostatic charge/discharge (GCD) cycles in the voltage range of −0.2 to 0.8 V, employing an Ag/AgCl reference electrode and 1 M H_2_SO_4_ as the electrolyte at different current densities of 0.5 to 4 A g^−1^. As depicted in Figure 4d–f (GCD curves at a current density of 1 A g^−1^), nearly linear charging/discharging profiles were observed for all samples, suggesting excellent energy storage capabilities using the PPy-based electrode materials [45]. The capacitance (*C*) can be determined from the charge–discharge curves using the following equation [8].
(1)$$C=\frac{I\times ∆t}{m\times ∆V}$$
where *I* is the charge/discharge current, Δ*t* is the discharge time, Δ*V* is the potential window (*V*), and M is the total mass of the active materials (g).

DMF PS@PPy20/30 exhibited the highest specific capacitance. The specific capacitances of DMF PS@PPy20 and DMF PS@PPy30, calculated from their charge/discharge curves, were 350 F/g and 303 F/g, respectively, at 1 A/g (Appendix A), which can be tuned even higher to 360 F/g and 362 F/g, respectively, using a current density of 0.5 A/g. The changes in the specific capacitance with current density are shown in Figure 4g–i. The discharge capacitances decreased at electric densities above 0.5 A g^−1^ due to the presence of irreversible pseudocapacitors. DMF PS@PPy30 retained 50% of its initial capacitance (from 362 to 181 F g^−1^), while DMF PS@PPy20 retained 39% (from 360 to 140 F g^−1^) of its initial capacitance when the current density ranged from 0.5 A/g to 4 A/g. The results demonstrate that DMF PS@PPy20 possesses a superior capacitance at low current density, while DMF PS@PPy30 possesses a superior rate performance at different charge–discharge rates, which will be discussed later.

The superior electrochemical properties observed for DMF PS@PPy20/30 can be attributed to several factors. First, the TGA and TEM results demonstrated that DMF PS@PPy20/30 had the lowest PS contents, indicating the efficient removal of nonconductive PS cores by DMF. Second, the thin sheath of the capsules reduces the ion diffusion length, facilitating rapid electrochemical reactions between the electrolyte and the PPy electrode. Finally, although XPS analysis indicated a decrease in the doping content of PPy due to DMF treatment, the efficient removal of nonconductive PS cores and the unique hollow nanostructure of the PPy shell still contributed to the excellent electrochemical performance of the samples.

Electrochemical impedance spectroscopy (EIS) was employed to assess the ion diffusion mechanism of the electrode materials. The Nyquist plots presented in Figure 5a–c exhibit distorted semicircles at high frequencies and nearly vertical linear spikes at low frequencies, which are indicative of capacitor materials. The inset in Figure 5a–c depicts the equivalent circuit comprising an equivalent series resistance (R_s_), charge transfer resistance (R_ct_), Warburg impedance (W), electric double-layer capacitance (C_dl_), and pseudocapacitance (C_F_). Appendix A lists the EIS characteristics of these materials. Among the samples, DMF PS@PPy20 exhibited the lowest Rs values, indicating high conductivity and capacitance. Additionally, compared with the control sample, DMF PS@PPy20 displayed the smallest semicircle diameter, suggesting that it had the lowest interfacial charge-transfer resistance (R_ct_). The sharp increase in the EIS curve indicated favorable capacitive behavior and rapid ion transport in the electrolyte (C_dl_, C_F_). Consequently, DMF PS@PPy20 exhibited the highest ionic and electrical conductivity, consistent with its good performance in CV and GCD tests. The higher conductivity of DMF PS@PPy20 can be attributed to its buckling structure, facilitation of electron and ion transfer, and decreased residual PS content. Overall, DMF PS@PPy20 demonstrated the best electrochemical performance.

The volumetric capacitance is another important parameter for supercapacitors. The volumetric capacitance of the PPy-based supercapacitors was calculated by multiplying the specific capacitance by the tap density of the PPy (Figure 5d–f) and is plotted in Figure 5g–i and Appendix A. Because of the low density of PPy (1.48 g/cm^3^), the volumetric capacitance of PPy-based electrodes is usually low. In addition, considering that the buckling structure can reduce the unnecessary void space in hollow spheres, the tap densities of all the samples were tested via NMR tubes and are listed in Figure 5d–f. The highest volumetric capacitance was 154 F/cm^3^ for DMF PS@PPy20 at a current density of 2.5 mA/cm^2^. The tap density of PS@PPy20 is the highest due to the shrinkage and compact structure. This discovery validates the significant influence of morphology on the electrochemical performance of PPy, as it dictates the electrode–electrolyte interphase. In essence, the structure of PPy varies according to its morphology and porosity, both of which can affect the electrochemical characteristics of PPy supercapacitors.

The charge storage kinetics of electrode materials are typically assessed through the analysis of cyclic voltammetry (CV) curves. By examining the CV data at different scan rates, the current contribution, including both surface-controlled capacitive current and diffusion-controlled current, can be deduced using the following equation [45]:(2)$$i=av^{b}$$

The peak current (*i*) at different scan rates (*v*) in the CV curves is analyzed to determine the kinetics of charge storage in electrode materials. The equation used incorporates adjustable parameters (*a* and *b*), with b calculated by plotting log(i) against log(v). The value of b allows us to differentiate whether the charge storage process is dominated by diffusion and exhibits battery-like behavior (b = 0.5) or if it is mainly governed by surface capacitance and displays capacitive behavior (b = 1). When 0.5 < b < 1, the charge transfer process involves a combination of pseudocapacitance and battery-type behavior [45]. In Figure 6a, the values of b for all PPy samples ranged from 0.5 to 0.7, suggesting that the PPy-derived current exhibited characteristics of both batteries and capacitors. Consequently, the total stored charge in PPy was primarily attributed to three processes: the nonfaradaic contribution of the double-layer effect, Faradaic intercalation during H-ion diffusion, and pseudocapacitance resulting from the capacitive process predominantly occurring at the surface during charge transfer. The DMF-treated samples showed higher b values, indicating an enhanced contribution from the double-layer capacitance, primarily due to an increase in the specific surface area.

The current at a constant potential can be expressed by the following equation, which represents two distinct current contributions [46]:(3)$$i\left(v\right)=k_{1}v+{k_{2}v}^{1/2}$$

Here, *k*_1_ and *k*_2_ are constants, *v* denotes the scan rate (mV s^−1^), and i(*V*) represents the current (A) under the constant potential (*V*). The values of k_1_ (slope) and k_2_ (intercept) can be determined by establishing a linear relationship between i(*V*) and v at different potentials.

The capacitance contribution can be quantified by calculating *k*_1_*v* and *k*_2_*v*^1/2^, which correspond to the current contributions from surface capacitive effects and diffusion-controlled intercalation processes, respectively. In Figure 6c,g–i, the blue region accounts for 32%, 35%, 36%, and 63%, respectively, of the integrated CV area (pink region), representing the diffusion-controlled capacitive contribution in PS@PPy20, PS@PPy30, DMF PS@PPy20, and DMF PS@PPy30. Figure 6i shows the capacity contribution reaching as high as 63% at a sweep rate of 30 mV s^−1^ for the DMF PS@PPy30 electrode. The results indicate that DMF PS@PPy20 predominantly exhibits pseudocapacitance resulting from fast redox reactions involving H^+^ intercalation, while DMF PS@PPy30 predominantly exhibits double-layer capacitance, described by equation [47].
(4)$${PPy}^{+}{(Cl}^{-})+H^{+}+e^{-}↔{PPy}^{0}(HCl)$$

The dominant contribution of surface capacitive effects in DMF PS@PPy30 explains its higher rate performance compared to that of DMF PS@PPy20 [48].

### 2.3. Semisolid-State Flexible Supercapacitor Device

To further evaluate the promising properties of the assembled device, a flexible supercapacitor was constructed using a DMF PS@PPy20 semisolid-state configuration [49]. Figure 7a showcases the cyclic voltammetry (CV) curves obtained for a two-electrode assembly. The supercapacitor was subjected to scanning within a voltage window of 0 V to 1 V at scan rates ranging from 5 to 100 mV s^−1^. Notably, all the curves maintain a rectangular shape, signifying excellent capacitive behavior and low internal resistance. Figure 7b presents the galvanostatic charge/discharge (GCD) behavior of the supercapacitor, where the charge curves exhibit near-symmetry with their discharge counterparts within the defined voltage window. This observation indicates the suitability of the supercapacitor for practical applications. Furthermore, a minor voltage drop is noticed as the current density increases from 0.5 A g^−1^ (289 F/g) to 4 A g^−1^ (141 F/g), indicating favorable capacitive performance. In Figure 7c, the Nyquist plot illustrates a small arc, which is attributed to low charge transfer resistance. Appendix A lists the EIS characteristics of DMF PS@PPy20-based flexible supercapacitors. The symmetrical supercapacitor demonstrates a low internal resistance of 5.46 Ω, along with a transfer resistance of 0.6 Ω. Figure 7d displays the Ragone plots, presenting the power and energy density determined through constant current charge/discharge at densities ranging from 0.5 A/g to4 A/g, as per the respective equations [26]:(5)$$E=\frac{1}{2}CV^{2}$$
where *E*, *C*, and *V* are the energy density (Wh/kg), the specific capacitance (F/g), and the cell voltage (V), respectively.
(6)P=E/∆t
where *P*, *E*, and Δ*t* are the power density (W/kg), the energy density (Wh/kg), and the discharge time (s), respectively.

In accordance with Figure 7d, the symmetric supercapacitor demonstrates a maximum energy density of 40 Wh/kg at a power density of 490 W/kg. Even at a high-power density of 4290 W/kg, the energy density remains at 21.0 Wh/kg [50]. It is worth noting that the energy density gradually decreases as the power density increases. This decrease can be attributed to the significant rise in both the large internal resistance and the ion diffusion resistance of the electrode material, stemming from the increased load of the active material. DMF PS@PPy20 shows higher energy density and power density than other recently reported PPy or PPy-based composites, which makes them competitive electrode materials for supercapacitors (Figure 7d) [51,52,53,54,55,56,57].

To illustrate the practical use of the PPy electrode, we conducted a demonstration where three symmetric supercapacitor devices were connected in series to power LEDs. Figure 7e displays the galvanostatic charge–discharge behavior of single, two, and three devices connected in this configuration. Figure 7f demonstrates the successful illumination of the LEDs using the fully charged supercapacitors. This confirms that the DMF PS@PPy20 composite possesses excellent supercapacitive characteristics and holds great potential as an electrode material for high-performance supercapacitors. It can effectively power real-life applications when integrated into such a system.

## 3. Experimental Section

### 3.1. Chemicals

Styrene monomer was purchased from Aladdin Reagent Co., Ltd. and purified using an inhibitor remover column. The purified monomer was stored at −5 °C. Pyrrole monomers from Shanghai Aladdin Bio-Chem Technology Co., Ltd. were redistilled and refrigerated at −5 °C. Analytical grade potassium persulfate (KPS), anhydrous ferric chloride (FeCl_3_), cyclohexane, toluene, tetrahydrofuran, chloroform, acetone, and N,N-dimethylformamide were purchased from Sinopharm Chemical Reagent Co., Ltd. Shanghai, China. Deionized water was used throughout the experiment.

### 3.2. Preparation of Polystyrene (PS) Particles

Monodisperse polystyrene (PS) particles were prepared according to previous work [58]. In the typical procedure, 140 mL of distilled water was added to 0.1 mmol of styrene monomer and stirred vigorously for approximately 20 min under a nitrogen atmosphere. Then, 10 mL of 0.023 mmol K_2_S_2_O_8_ was gradually added under stirring, and the mixture was maintained at 80 °C for 24 h. The resulting PS particles were centrifuged and washed multiple times with distilled water.

### 3.3. Preparation of Polystyrene@polypyrrole (PS@PPy) Core–Shell Nanospheres and Hollow PPy Spheres

The synthesis procedure involved the following steps: 0.5 g of PS particles were dispersed in 30 mL of deionized water. The mixture was then deoxygenated for 30 min, after which an appropriate volume of pyrrole monomer was added. After 10 min, 5 mL of an aqueous solution of FeCl_3_·6H_2_O was slowly added dropwise to the reaction mixture under a nitrogen atmosphere, followed by stirring. The polymerization reaction was carried out at room temperature for 24 h. Upon completion, the resulting products were washed with water and alcohol through centrifugation cycles and then dried in an oven at 65 °C. The obtained PS@PPy core–shell nanoparticles served as precursors for the fabrication of hollow PPy spheres. For this purpose, six organic solvents, namely tetrahydrofuran (THF), toluene, chloroform, N, N-dimethylformamide (DMF), acetone, and cyclohexane, were chosen to dissolve the PS and obtain hollow PPy spheres. PS@PPy powder and the respective organic solvents (1:10 weight ratio) were stirred for 24 h, washed with the corresponding organic solvent through centrifugation cycles, and dried in an oven at 65 °C under vacuum.

### 3.4. Characterization

Scanning electron microscopy (SEM, Quanta 400 FEG, FEI Amsterdam, The Netherlands) and transmission electron microscopy (TEM, FEI Tecani G2 F20 S-TWIN, FEI Amsterdam, The Netherlands) were used to investigate the morphology of the samples. Fourier transform infrared (FTIR) spectra were obtained for the samples with a Perkin-Elmer FTIR spectrometer (Spectrum System 2000, Perkin-Elmer, Waltham, MA, USA) using potassium bromide pellets. X-ray photoelectron spectroscopy (XPS) measurements of the powders on a glass substrate were performed using electron spectroscopy for chemical analysis (ESCA) instrument (VG Multilab 2000, Thermo Fisher Scientific, Waltham, MA, USA). High-resolution scans with good signal ratios were obtained for C 1s, N 1s, and O 1s. All the spectra were recorded under ambient conditions. A thermogravimetric analyzer (TGA Q50, TA Instruments, New Castle, DE, USA) was used to examine the thermal degradation properties of the prepared core–shell particles. The sample weight was 10 mg. The experimental run was performed from 20 to 800 °C at a heating rate of 10 °C min^−1^ in a nitrogen atmosphere with a gas flow rate of 30 mL min^−1^.

### 3.5. Electrochemical Characterization

#### 3.5.1. Three-Electrode System

The carbon cloth (1 cm × 2 cm), used as a current collector, was washed with deionized water and ethanol to remove impurities. This was followed by treatment with concentrated HNO_3_ for 8 h at room temperature. The treated carbon cloth was then washed with deionized water and ethanol again and dried in a vacuum oven at 60 °C for 30 min.

The working electrode was made by mixing 80 wt% PPy, 10 wt% acetylene black, and 10 wt% poly(vinylidene fluoride) in N-methyl-2-pyrrolidone and the slurry was coated onto a carbon cloth (1 cm × 2 cm) current collector and dried at 60 °C for 8 h to evaporate the solvent. The mass loading of materials on the carbon cloth was obtained by measuring the weight difference before and after coating using a microbalance. The mass loaded (PPy) on the carbon cloth was 1 mg/cm^2^.

A three-electrode test system was used for cyclic voltammetry (CV) and galvanostatic charge–discharge (GCD) measurements by applying 1 M H_2_SO_4_ as the electrolyte and a platinum plate and Ag/AgCl (3 M KCl) as the counter and reference electrodes, respectively, via an electrochemical workstation (CHI 660D) with a potential range from −0.2 to 0.8 V. Electrochemical impedance spectroscopy (EIS) was performed with a three-electrode system in the frequency range from 100 kHz to 0.01 Hz (AC voltage, 5 mV).

#### 3.5.2. Flexible Two-Electrode Cell

The solid electrolyte (PVA-H_2_SO_4_ gel) was prepared by adding 5 g of PVA to 50 mL aqueous solutions and stirring at 85 ℃ until the PVA was completely dissolved, after which 5 mL of H_2_SO_4_ was added. The flexible electrode was prepared with carbon cloth (1 cm × 1 cm) as the matrix, on which a mixed slurry of active material, acetylene black, and PVDF was coated at a mass ratio of 80:10:10. Two identical flexible electrodes were assembled with the as-prepared PVA-H_2_SO_4_ solid gel electrolyte, and finally fixed with a plastic film to obtain a symmetric supercapacitor device.

## 4. Conclusions

In summary, we have demonstrated a universal strategy for preparing hollow PPy nanoparticles via the use of an organic solvent to etch a hard template. DMF was shown to be the appropriate solvent for the preparation of hollow PPy nanoparticles due to its favorable polarity, vapor pressure, and solubility parameters. The morphology of PPy was found to affect electrochemical performance. The specific capacitances of DMF PS@PPy20 and DMF PS@PPy30, calculated from their charge/discharge curves, were 350 F/g and 303 F/g at 1 A/g, respectively. Furthermore, the symmetrical supercapacitor constructed using DMF PS@PPy20 displayed a high energy density of 40 Wh/kg at 490 W/kg, maintaining an energy density of 21.0 Wh/kg even at an elevated power density of 4290 W/kg. Consequently, DMF PS@PPy20 demonstrates promising potential as an active material for high-performance supercapacitors. These findings may provide new insights into both the synthesis and energy storage mechanisms of PPy nanoparticles.

## Figures and Tables

**Figure 1 molecules-29-02331-f001:**
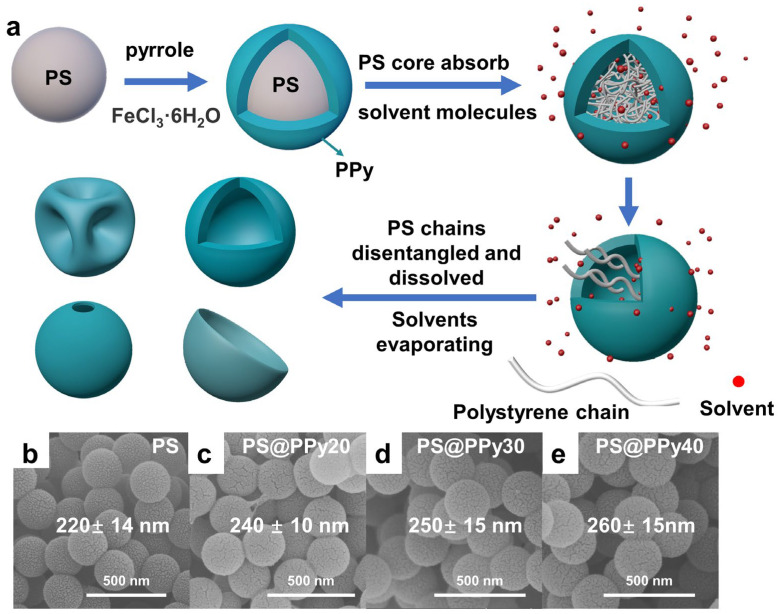
(**a**) Schematic illustration of the synthesis procedure of hollow PPy nanoparticles with an organic solvent etching strategy; SEM images of (**b**) PS, (**c**) PS@PPy20, (**d**) PS@PPy30, and (**e**) PS@PPy40.

**Figure 2 molecules-29-02331-f002:**
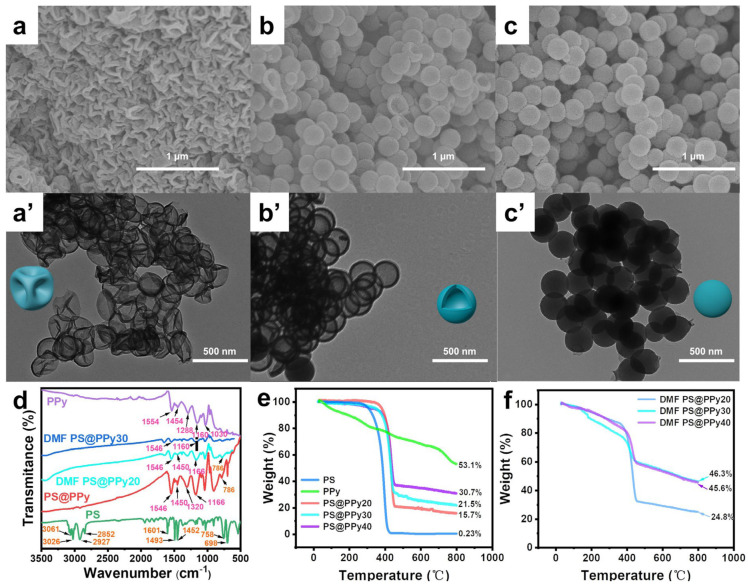
SEM and corresponding TEM images of (**a**,**a’**) DMF PS@PPy20, (**b**,**b’**) DMF PS@PPy30, and (**c**,**c’**) DMF PS@PPy40; (**d**) FTIR spectra of PS, PPy, PS@PPy, and DMF PS@PPy; TGA traces of PS, PPy, PS@PPy (**e**), DMF PS@PPy20, DMF PS@PPy30, and DMF PS@PPy40 (**f**).

**Figure 3 molecules-29-02331-f003:**
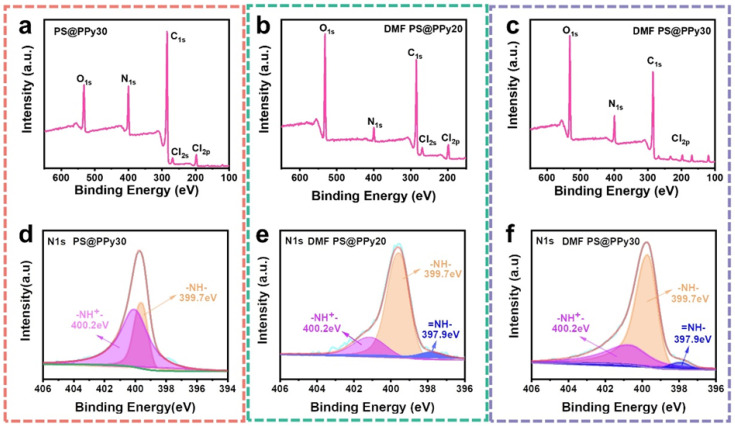
XPS spectra of (**a**) PS@PPy30, (**b**) DMF PS@PPy20, and (**c**) DMF PS@PPy30. N_1s_ spectra of (**d**) PS@PPy30, (**e**) DMF PS@PPy20, and (**f**) DMF PS@PPy30.

**Figure 4 molecules-29-02331-f004:**
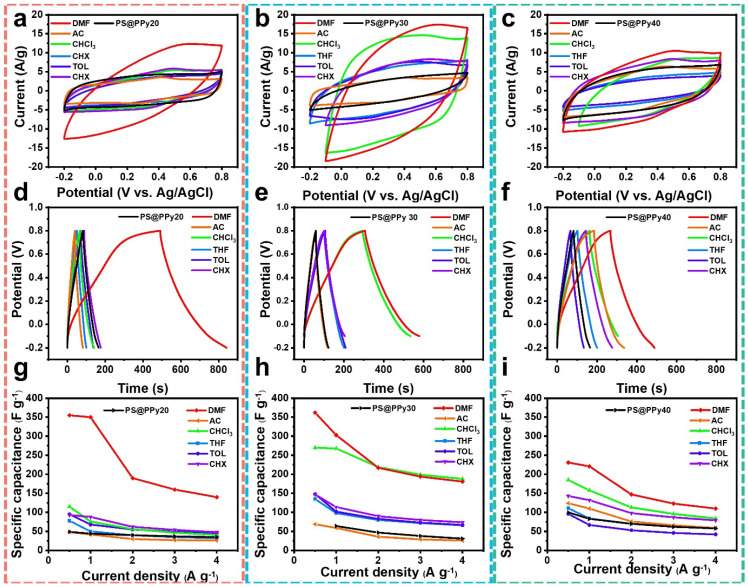
Capacitive performance of the organic solvent−treated PS@PPy20/30/40 electrodes. (**a**–**c**) CV curves at a scan rate of 100 mV s^−1^; (**d**–**f**) GCD curve at a current density of 1 A g^−1^; (**g**–**i**) specific capacitance changes with different current densities.

**Figure 5 molecules-29-02331-f005:**
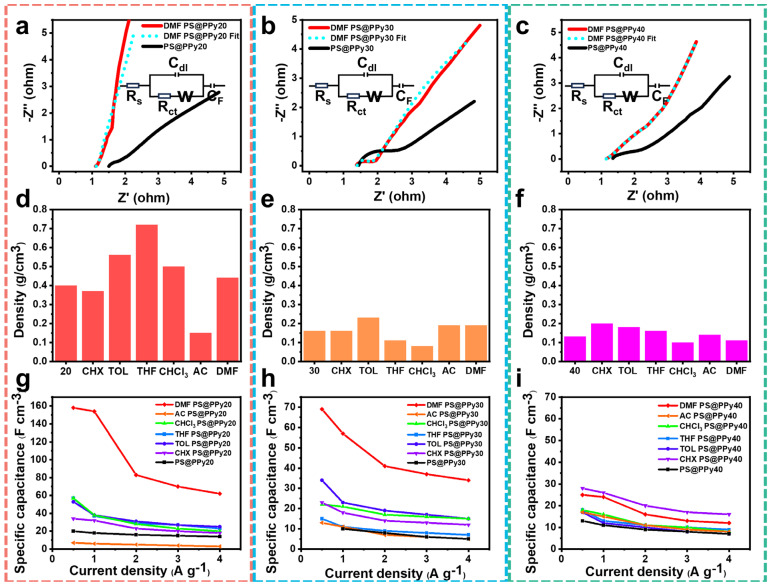
Nyquist plots of DMF PS@PPy (**a**–**c**). The insets show the equivalent circuit. Tapping density of organic solvent−-treated PS@PPy20 (**d**), PS@PPy30 (**e**), and PS@PPy40 (**f**). Volumetric capacity of organic solvent-treated PS@PPy20 (**g**), PS@PPy30 (**h**), and PS@PPy40 (**i**).

**Figure 6 molecules-29-02331-f006:**
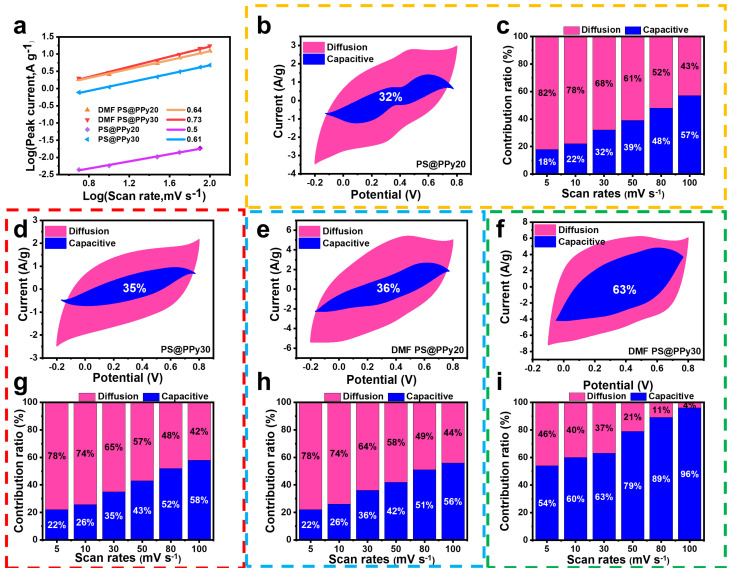
(**a**) Linear relationship between the logarithm of the peak current and the logarithm of the scan rate for the PPy−-based electrode. Capacity contribution of PS@PPy20 (**b**,**c**), PS@PPy30 (**d**,**g**), DMF PS@PPy20 (**e**,**h**), and DMF PS@PPy30 (**f**,**i**) at different scan rates.

**Figure 7 molecules-29-02331-f007:**
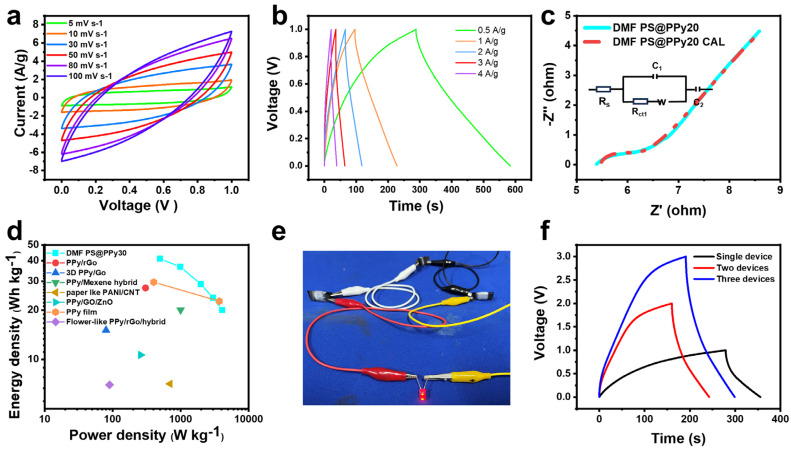
CV (**a**), GCD (**b**), and Nyquist (**c**) plots of DMF PS@PPy−-based flexible supercapacitors. (**d**) Ragone plots of DMF PS@PPy20 and the values of recently reported PPy or PPy-based composites for comparison. (**e**) GCD curves of one, two, and three supercapacitors connected in series at 1 A/g. (**f**) DMF PS@PPy20 capacitive device diagram.

**Table 1 molecules-29-02331-t001:** Solvent parameters and corresponding morphologies of hollow PPy nanoparticles.

Solvent	Polarity	Solubility Parameter cal^0.5^ cm^−1.5^	Boiling Point (°C)	Vapor Pressure (kPa)	Morphology (PS@PPy20)	Morphology (PS@PPy30)	Morphology (PS@PPy40)
Cyclohexane	0.1	7.3	80.7	13.33	intact	intact	intact
Toluene	2.4	8.9	110.6	4.00	collapsed	buckling	intact
Tetrahydrofuran	4.2	9.5	66.0	23.46	collapsed	buckling	intact
Chloroform	4.4	9.21	61.2	26.45	collapsed	buckling	buckling
Acetone	5.4	9.8	56.5	30.56	intact	hollow	buckling
N,N-dimethylformamide	6.4	12.1	153.0	0.50	buckling	hollow	intact

**Table 2 molecules-29-02331-t002:** Elemental content and surface composition of nitrogen.

	C	N	O	Cl	-NH-	-NH^+^-	-NH=
PS@PPy30	71.6%	15%	10.9%	2.5%	31.5%	68.5%	/
DMF PS@PPy20	62%	6.1%	27.2%	4.7%	67.6%	25.4%	7%
DMF PS@PPy30	59%	11.5%	28%	1.5%	68%	28%	4%

## Data Availability

Data are contained within the article and Appendix A.

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
