# Peer review of "Fabrication of Polypyrrole Hollow Nanospheres by Hard-Template Method for Supercapacitor Electrode Material"

_molecules, 2024, doi:10.3390/molecules29102331_

Round 1

Reviewer 1 Report

Comments and Suggestions for Authors

In the attached file, you will find the review report

Author Response

Please see the attachment。

Reviewer 2 Report

Comments and Suggestions for Authors

In the manuscript, the authors present a study on polystyrene-polypyrrole core-shell nanoparticles with various polypyrrole thickness which were synthesized according to the hard template method. The polystyrene cores were etched using different solvents such as cyclohexane, tetrahydrofuran, trichloromethane, toluene, acetone and N,N-dimethylformamide which were reported earlier for PS core dissolution. The authors investigate nanostructure and evaluate electrochemical properties of the hollow polypyrrole shell which is obtained after the process of core etching. In spite of the fact that the paper does not show a breakthrough, the novelty of the study is related to the authors’ claim that the dissolution process, the corresponding structure of the PPy shell, and its impact on the electrochemical properties, have not been extensively studied.

The article is clear and relevant to the scientific field, the experimental part is well written, and the study is correctly designed. From my opinion, the manuscript is scientifically sound, and the experimental design is appropriate. The cited references are relevant to the field. The figures show the data properly, and they are easy to interpret and understand for readers. The conclusions are consistent and supported by the obtained research data.  

The manuscript could be interesting for scientists working in the fields of polymer science, electrochemistry, supercapacitors and some related topics. However, a few issues must be addressed. From my opinion, the article is within the scope of Molecules and may be published in this journal after minor revision.

  1. The strong point of the study is an attempt to use the solubility parameters in order to explain the research findings. However, the authors make only one conclusion, saying that these solvents in principle could be used for the PS dissolution. Can the authors discuss more the solubility parameters and try to explain how the best solvent is related to its solubility parameter and the solubility parameter of PS? How does this relate to the polarity?
  2. The authors claim that “PPy nanospheres synthesized with DMF demonstrated superior electrochemical properties compared to those prepared with other solvents, attributed to their highly effective PS removal efficiency, increased specific surface area…” What is about the specific surface area of the resulting samples? Was it measured or assumed? Can the authors confirm the increase of the specific surface area?
  3. For this type of paper, I would suggest to mention some numerical data such as specific capacitance and/or energy density in the abstract.
  4. Line 66. “…between positive charged PPy and negative charged PS particles…” Usually it is said “…positively charged…” and “…negatively charged…”.
  5. Line 67. Figure 1a. According to the text of the manuscript, the core consists of polystyrene (PS), not polypyrrole (PPy). The image on the second step shows the core of PPy, which is the shell in fact. Please, clearly label the core and shell in the Figure 1а. 
  6. Line 52: trichloromethane is usually called chloroform.
  7. Line 192: The authors mention “to 100 mV s-1 c [38].” What is “c”?
  8. Line 371: The authors claim that “Monodisperse polystyrene (PS) particles were prepared according to previous work.” Which work do the authors mean?

Reviewer 3 Report

Comments and Suggestions for Authors

Review Comments:

The manuscript “Fabrication of polypyrrole hollow nanospheres by 2 hard-template method for supercapacitor electrode material” reported the synthesis method 25 and energy storage of PPy nanoparticles.. Overall, the authors characterized the structure of the material well and comprehensively, but some issues in this paper would need to be addressed before being accepted into publication: Minor revision

1. Why authors not following consistence in the potential window of CV and GCD measurements?.

2 There are some grammatical and typo errors please correct them. Some places, wording of the sentences are incompatible, please correct them.

3. please correct the Figure 5 description symbols.

4. The charging and discharge time of device looks huge difference, why? And the current density values mentioned in the Figure 7b and in the matter are different, please recheck and correct them.

5. “Monodisperse polystyrene (PS) particles were prepared according to previous work” give the reference.

6. Introduction looks very less, please compare with the previous studies on PPy to improve the introduction part.

Comments on the Quality of English Language

Minor changes are needed.

Round 2

Reviewer 1 Report

Comments and Suggestions for Authors

The minor and major corrections were well executed by the authors. In the calculation of Eg through UV-vis spectroscopy, the authors could have engaged in a more extensive discussion, as the observed results are quite intriguing. For the UV-vis spectra, a good level of doping is evidenced by the continuous absorption over 600 nm, which is consistent with the obtained optical band gap (approximately 1.2-1.3, and also in the voltammograms of cyclic voltammetry). Due to the high doping level, what is strictly being determined is not the band gap but the transition from the valence band to the first intermediate conduction band. This value is small, which indicates a high conductive capacity of the polymer. This phenomenon could have been further discussed by the authors. Although the intriguing UV-vis results have not been discussed in depth, the manuscript is of excellent quality.